# Blast Fragment Impact of Angle-Ply Composite Structures for Buildings Wall Protection

**Daniel Barros** [1,2,*], **Carlos Mota** [1,2], **João Bessa** [1,2], **Fernando Cunha** [1,2], **Pedro Rosa** [3] and **Raul Fangueiro** [1,2]

1  Fibrenamics Association, 4800-058 Guimarães, Portugal; cmota@tecminho.uminho.pt (C.M.); joaobessa@fibrenamics.com (J.B.); fernandocunha@det.uminho.pt (F.C.); rfangueiro@dem.uminho.pt (R.F.)
2  Centre for Textile Science and Technology, University of Minho, 4800-058 Guimarães, Portugal
3  IDMEC, Instituto Superior Técnico, University of Lisbon, 1049-001 Lisbon, Portugal; pedro.rosa@tecnico.ulisboa.pt
*  Correspondence: danielbarros@fibrenamics.com

**Abstract:** This paper investigates the fragment performance of several composite panels for attaching to the inside walls of a building structure. These panels were developed using different types of fiber woven fabrics (W0, W90) combined with distinct layers orientations (angle-ply effect) of L0/0 and L0/15. Aramid, E-glass, and S-glass fiber fabrics impregnated with thermosetting epoxy resin, and a prepreg of Ultra High Molecular Weight Polyethylene (HB24) were employed. The panels are subjected to ballistic impact using different fragments under impact velocities in the range of 120 to 420 m/s. In order to measure the energy absorbed by the ballistic panels, the impact velocity and the residual velocity of the fragment were measured with laser chronographs placed before and after the laminated test specimens. The paper demonstrates quantitatively that the angle-ply laminates produced using L0/15 woven fabric orientation presented a higher impact energy absorption, promoting higher reductions on the fragment residual velocity compared to the L0/0 orientations. The laminates produced using UHMWPE fibers (HB24) presented better ballistic properties compared to the other fibers. Furthermore, it was noted that the energy dissipation rate is linearly correlated with the impact velocity and is independent of the fragment geometry.

**Keywords:** blast; fragment; impact; building; angle-ply; composite; glass; aramid; UHMWPE; fibers

## 1. Introduction

Nowadays, the occurrence of terrorist attacks is frequent, as well as the presence of wars in some countries. These events take place mainly in cities, causing damage to their buildings, jeopardizing their structure, and, in the worst scenarios, risking the integrity of their inhabitants. However, most existing buildings were not designed to withstand blast loading [1,2]. Most casualties and injuries sustained during external explosion are not caused by the bomb detonation but because of the wall fragments that can be propelled at high velocities by the blast. Therefore, it is necessary to create defense mechanisms capable of resisting the impact of these fragments projected from the wall. The resistance of a wall to blast loads can be enhanced by increasing the mass and ductility of the wall with additional reinforcement materials, as in the case of fiber composites [3,4].

Ballistic/fragment protections are mainly composed using steel and conventional composite panels. However, due to the associated heavy weights and corrosion problems, it is necessary to develop improved new solutions with the same levels of protection but with less weight. The continuous evolution of the materials, with a special focus on composite materials, allows for the development of better combinations of materials in order to get lightweight solutions without penalizing the ballistic protection levels [5–8].

During the last decades, various polymer composite combinations have been explored for protective applications by processing different kinds of fibers, such as aramid, E-glass, and UHMWPE (ultra-high molecular weight polyethylene), due to their high specific energy

absorption and dissipation under high energy impacts. For better performances, these types of fibers are impregnated with specific polymeric matrixes, such as thermosetting resins, to improve the ballistic performance of the solutions [9–13]. To produce these type of application, woven fabrics are normally used to promote the ballistic protection against projectiles. However, they may not be efficient against small fragments with a size similar to the yarn-to-yarn spacing of the used woven. It is possible in such cases that the fragment may slip through the woven yarns and the impact energy is not transferred to the fabric [14–16]. Thus, the number of woven layers and the layer orientation must be optimized in order to promote the energy dissipation [17].

During a ballistic/fragment impact, there are several variables that can influence the ballistic protection, such as the type of fiber, the polymeric matrix, the resin volume fraction, the woven pattern, and the woven fibers' layer orientation [18,19]. Thus, during a ballistic impact, a complex stress distribution is verified due a simultaneous combination of mechanical loads, namely fibers subject to tensile, flexure, compressing, and shear forces, caused by the interaction of the projectile/fragment with the composite laminate [20–22]. Figure 1 presents a schematic illustration of this complex interaction between the projectile and the protective composite panel.

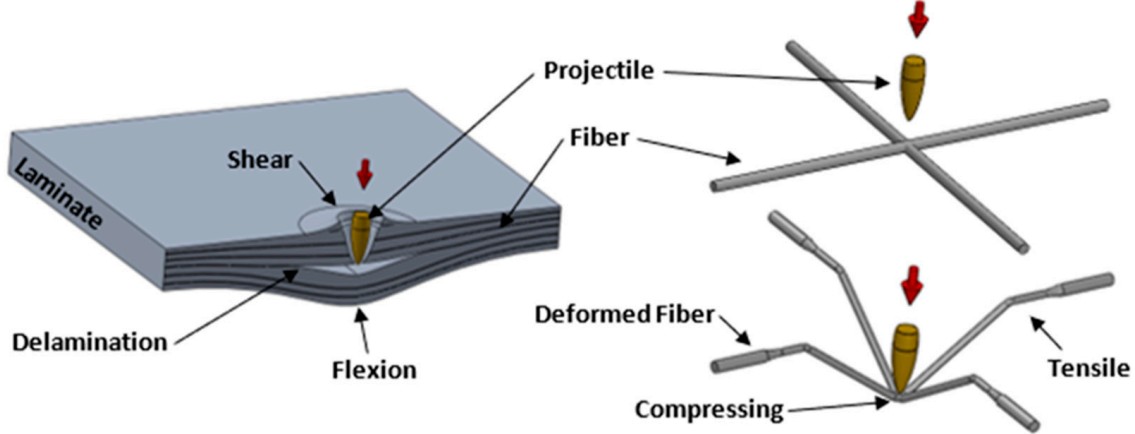

**Figure 1.** Some mechanisms associated with a ballistic impact.

The fibers have the function of reinforcing the capacity to absorb and dissipate the projectile kinetic energy, but the role of the matrix is also paramount, by restricting the lateral motion of the fibers, helping with the dissipation of the impact energy to the successive layers during an impact, increasing the overall dissipation absorption capability of the laminate [23,24].

When a projectile hits a composite fiber target, due to its small dimension, it has contact with only a few strands of yarns, and, in a study made by Cunnif [15,25], the yarns that have contact with the projectile have been named principal yarns, while the others that are not in direct contact were referred to as secondary yarns. So, during a ballistic impact, the projectile creates a transverse deflection in the principal yarns, and that deflection generates a longitudinal stress wave in the secondary yarns. When the transverse deflection reaches its limit, the principal yarns fail at their breaking strain, and the projectile continues to the next yarn. The principal yarns are the ones that absorb the majority of the projectile kinetic energy, and the secondary yarns are often not fully stressed before woven failure. So, the limited involvement of secondary yarns in energy absorption is promoted by the inherent orthogonal nature of plain weave structure. Thus, it is important to design isotropic or quasi-isotropic compliant structures using high-strength, high-modulus fibers as ballistic armor material [26,27].

Currently, new solutions continue to be investigated to improve the impact resistance performance of composite panels, such as the application of woven fabrics with different orientations along the multilayer composite. To produce ballistic composite laminates,

typically used are layers of woven fabrics (W0, W90), all of them with the same fiber orientation, (L0/L0), since this is the easier methodology [28]. However, it is related in some studies that different layer orientations along the multilayer composite can increase the impact resistance performance, due to the increase of the projectile energy dissipation, making it easier to stop the projectile. This behavior can be justified by the dispersion of the fibers' yarn orientation, contributing to the energy dissipation [29]. This way, there is a reduction in the space between yarn that the projectile can slip through, and there is a massive interaction between the principal and secondary yarns in all the multilayer composite, improving the ballistic resistance of a target.

The present paper offers a comprehensive assessment of the high velocity impact performance of composites developed using different (W0, W90) types of fiber-woven fabrics developed using different layer orientations (reproducing an angle-ply effect). The angle-ply structures (0° and 15° layer orientations) used were based on a previous work, in which the impact of energy absorption of thermoset and thermoplastic composites was studied [30,31].

## 2. Experimental Procedures

### 2.1. Materials

To produce the laminates, the following were used: taffeta woven fabrics (W0, W90) of aramid (80 g/m$^2$, warp/weft 159 dtex) and E-glass (300 g/m$^2$, warp/weft 204 dtex), purchased from Castro Composites, Spain; a prepreg of a taffeta woven fabric of S-glass (815 g/m$^2$, with 33% epoxy resin content, warp/weft ~2400 dtex), purchased from Gurit, Germany; and prepreg of taffeta UHMWPE (HB24) ($\pm$240–271 g/m$^2$, four sublayers bidirectional orientated (90°)), purchased from Dyneema$^\circledR$. The properties of these fibers are presented in Table 1. The epoxy resin used to impregnate the woven fibers was the SR GreenPoxy 33 and the hardener SZ 8525, which were purchased from Rebelco, Portugal.

**Table 1.** Properties of the fibers used.

| Properties | Symbol | Units | E-Glass | S-Glass | Aramid | UHMWPE |
|---|---|---|---|---|---|---|
| Density | $\rho$ | kg/m$^3$ | 2119 | 2009 | 1320 | 960 |
| Areal density | | g/m$^2$ | 300 | 815 | 80 | 255 |
| Young's Modulus | E1 | GPa | 34.62 | 24.76 | 28.89 | 34.25 |
| | E2 | | 34.62 | 24.67 | 28.89 | 1.2 |
| Poisson's ratio | $\upsilon12$ | | 0.29 | 0.18 | 0.18 | 0.013 |
| Modulus of rigidity | G21 | GPa | 5 | 2.79 | 2.12 | 0.548 |
| | G22 | | 5.5 | 2.79 | 2.12 | 0.548 |
| Tensile strength | $\sigma$T1 | MPa | 363 | 471.79 | 307 | 1250 |
| | $\sigma$T2 | | 363 | 471.79 | 307 | 1250 |
| Compressive strength | $\sigma$C1 | MPa | 409 | - | 94.39 | 1900 |
| | $\sigma$C2 | | 92.2 | - | 112.66 | 1900 |
| Breaking strength | $\sigma$r1 | MPa | 355 | 342.87 | 299.17 | - |
| | $\sigma$r2 | | 355 | 342.87 | 299.17 | - |
| Elongation at break | $\varepsilon$ | % | 1.53 | 2.76 | 1.5 | - |

### 2.2. Sample Development

The woven fabrics (W0, W90) were cut into rectangular specimens of 350 × 250 mm, in different woven orientations in order to obtain layers with an orientation of 0° and 15°, allowing us to produce the angle-ply laminates. After that, each layer of aramid and E-glass was manually impregnated with the epoxy resin. In the case of the prepregs, the S-glass were purchased already impregnated with an epoxy matrix, and the UHMWPE was purchased with a polyurethane (PUR) matrix, so it was not necessary to impregnate them. After the cutting and impregnation of the pieces for each fiber, the multilayer composites were developed, with the deposition of each fiber layer with 0° and 15° orientations, in order to produce an angle-ply structure, as presented in Figure 2.

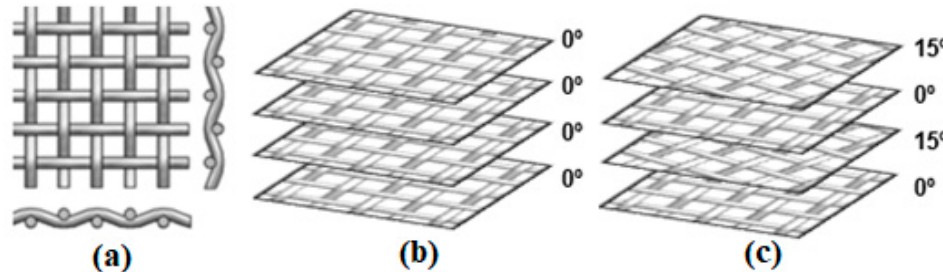

**Figure 2.** Angle-ply orientations in study: (**a**) Fiber orientations of the used woven fabrics (W0, W90), (**b**) Depositions of layers with L0/L0, and (**c**) with L0/L15.

After the preparation of the several materials, each sample was processed on a hot compression, molding pressing machine in order to cure the epoxy resin or melt the prepreg and compact the fibers. To process and cure each laminate, it was applied around $11 \pm 2$ bar and pressed at 95 °C (and 130 °C for the HB24 laminate) for 25 min. All samples produced are presented in Table 2, where the resin mass fraction, the thickness of the laminate composites, and number of layers of each fiber used are shown.

**Table 2.** Produced samples.

| Sample | Orientation | Reference | Resin Mass Fraction (%) | Laminate Thickness (mm) | Number of Layers |
|---|---|---|---|---|---|
| Aramid | L0/0 | A0/0 | 29 | 1.14 | 12 |
| | L0/15 | A0/15 | 28 | 1.09 | 12 |
| HB 24 | L0/0 | HB0/0 | - | 1.16 | 4 |
| | L0/15 | HB0/15 | - | 1.10 | 4 |
| E-glass | L0/15 | E0/15 | 31 | 1.13 | 6 |
| S-glass | L0/15 | S0/15 | 33 | 1.22 | 2 |

With the produced laminates, the ballistic impact tests were conducted.

### 2.3. Impact Testing

To simulate the impact of a fragment from an explosion, low-speed ballistic impact tests were carried out in the Terminal Ballistics Laboratory of the Instituto Superior Técnico, Portugal, using a specially designed gas gun and a safety chamber apparatus. This apparatus, schematically shown in Figure 3, makes use of a set of chrono-graphs to measure the impact velocity (before hitting the specimen) and the residual velocity (after passing through the specimen) of the fragments for all test conditions. The gas gun consists of a pressure vessel that allows precise and continuous control of the amount of energy released during compression for a given air pressure value. A pneumatic trigger valve allows the stored air volume to flow through a barrel, converting pneumatic energy into kinetic energy and, thus, accelerating the projectile/fragment until the desired impact velocity is reached. An adequate gun barrel was selected according to each specific type of projectile/fragment (geometry, material, and caliber). Three different types of simulative fragments have been used, as show in Figure 4. Simulated fragments have calibers of 5.5 mm (HL and PRP) and 7.9 mm (E), and, thus, it was necessary to use two different barrels with inner diameters of 5.56 mm and 8 mm, respectively. The first two types have been tested in the range of 24 J and 40 J, while the third one was in the range of 160 J. Laminate panels were produced to a maximum of 1 mm thickness in order to allow passing shot testing in the whole impact energy range. To compare the ballistic performance of the developed panels, some ballistics tests were performed according to the MIL-STD-662F standard. At least five subtests were performed per test condition. Impact energy and residual energy were calculated from their corresponding impact velocity and residual velocity. Energy absorbed by the composite specimen was estimated considering the difference between the two energy values above. The total absorbed energy value was considered as the main parameter to assess "ballistic" performance.

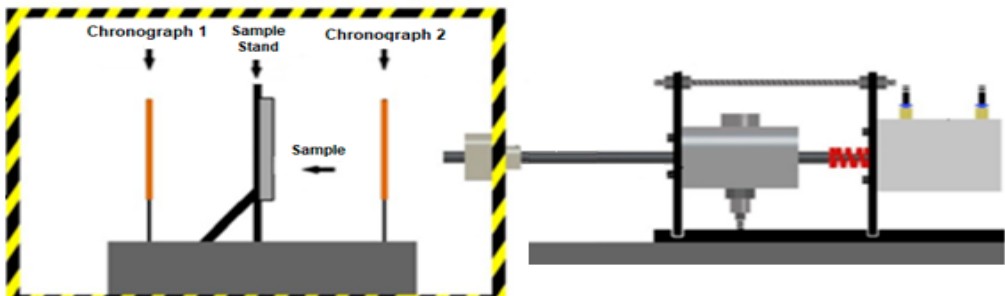

**Figure 3.** Schematic representation of the specially designed ballistic impact apparatus using a gun with variable gas power.

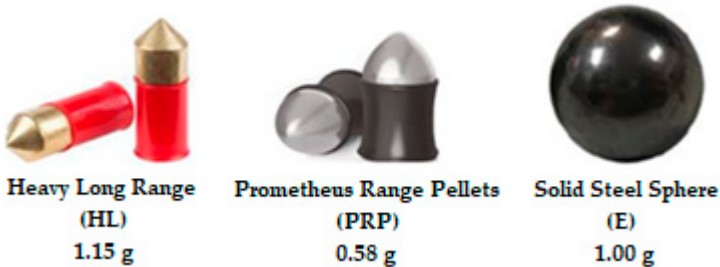

**Figure 4.** Different fragments used for the impact tests.

## 3. Results and Discussion

This section of the paper is structured in three parts. The first two parts are focused on the influence of the fragment type and the effect of the angle-ply structure on the residual velocity, respectively. The last section presents the impact absorption capability of the different fabric fibers used.

### 3.1. Impact Tests—The Effect of Fragments

Three different types of projectiles were used in order to evaluate the influence of the geometry/shape of the fragment in the laminate's impact absorption capability. For this, the Heavy Long (HL), the Prometheus Range Pellets (PRP), and the solid steel sphere (E) fragments were fired into an aramid laminate (A0/0), The aramid L0/0 was selected because its higher number of layers in its 1mm thickness composition highlight the influence of the fragment type on the impact resistance. Figure 5 shows a typical plot of the residual/final velocity as a function of the impact velocity retrieved from the experimental tests using different fragment types.

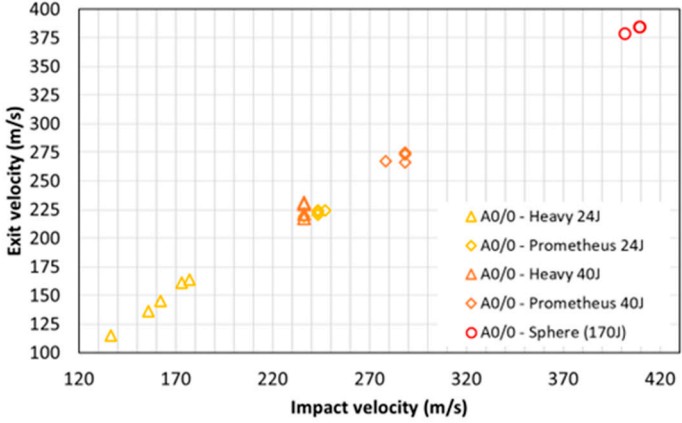

**Figure 5.** Residual velocity as a function of impact velocity for different types of fragments impacting aramid L0/0 laminate.

This figure allows two different types of conclusions to be drawn. First, it shows that the residual velocity is directly proportional to the impact velocity applied on the targeted specimens. Second, it reveals that residual velocity is not significantly influenced by the fragment type. In fact, the deviation of the residual velocity observed over the full range of test conditions is very small. A nonlinear correlation would be expected for a wider range of the impact velocity.

From the graph, it is possible to identify the different types of fragments used by the symbols: triangle, diamonds, and circles.

Figure 6 presents the front face (entry) and back face (exit) for some fragment types and test conditions, and Table 3 presents the hole diameters on the front face and the damage area on the back face, promoted by the impact of each type of fragment, measured using the Image J 1.43u software.

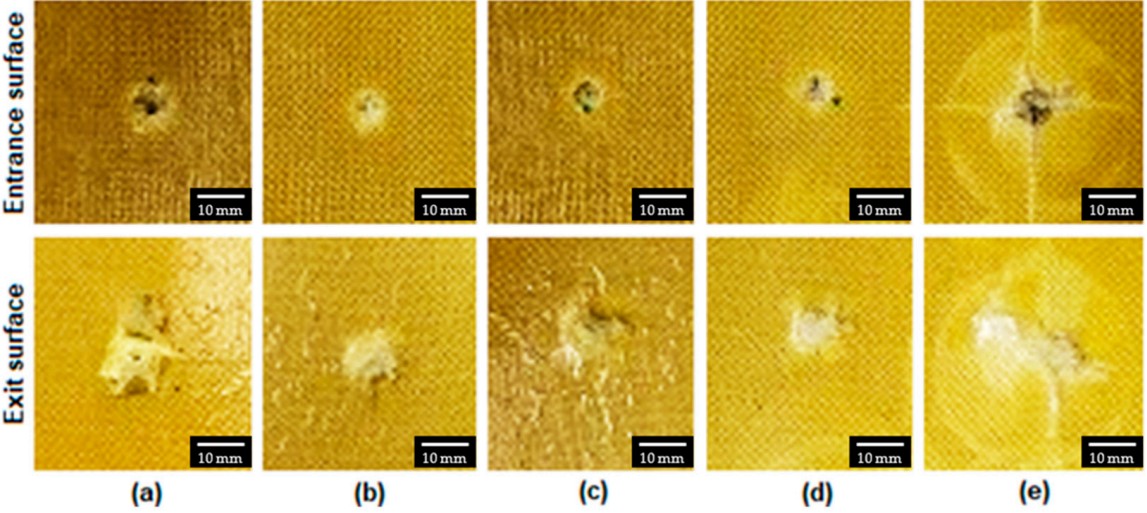

**Figure 6.** Front face (entrance surface) and back face (exist surface) of the aramid L0/0 laminate after fragment impact: Heavy Long (**a**) 24 J and (**b**) 40 J, Prometheus Range Pellets (**c**) 24 J and (**d**) 40 J, and solid steel sphere (**e**) 160 J.

**Table 3.** Sample damage evaluation after each fragment's impact.

| Sample | Projectile | Projectile Diameter (mm) | Impact Energy (J) | Front Face Impact Hole (mm$^2$) | Back Face Areal Damage (mm$^2$) |
|---|---|---|---|---|---|
| Aramid L0/0 | HL | 5.5 | 24 | 55.2 ± 0.03 | 150.1 ± 2.25 |
| | | | 40 | 57.2 ± 0.01 | 113.8 ± 0.54 |
| | PRP | 5.5 | 24 | 56.2 ± 0.01 | 115.7 ± 1.43 |
| | | | 40 | 55.2 ± 0.02 | 100.0 ± 0.62 |
| | Solid Sphere | 7.9 | 160 | 78.9 ± 0.01 | 1500.2 ± 9.71 |

Fragment entrance has an entry hole diameter proportional to the fragment size. As expected, 7.9 mm solid spheres provided a greater hole area (around 79 mm$^2$). Visual inspection of the back face also showed a higher damaged area for the solid steel sphere, 1500 mm$^2$. This can be explained by the diameter of the fragment but also because of the higher impact energy employed (160 J). Regarding the other fragment types, the entry hole area and the damage area were identical, mainly because of the narrow impact velocity range. Thus, the influence of the fragment geometry/shape has a minimal influence on the ballistic performance in this impact energy range (20 to 40 J). The areal damage measured on the back face of the sample for the HL and PRP fragments was around 100 and 150 mm$^2$.

### 3.2. Impact Tests—The Effect of the Angle-Ply Structure

First was tested the different types of fragments; then was studied the effect of applying an angle-ply structure to the laminate, trying to improve the impact resistance. For this study, laminates made of aramid fibers were selected because these materials have been studied over the years for this kind of application. In comparison, the HB24 fibers were selected because this type of fiber is a new trend for this type of application due to its impact resistance properties and low density, compared to the others. So, to evaluate the influence of angle-ply structures in the impact resistance, the aramid and HB24 laminates, A0/0, A0/15, HB0/0, and HB0/15, were tested using the PRP (40 J) and E (160 J) fragments. The impact and exit fragment velocities are presented in Figure 7.

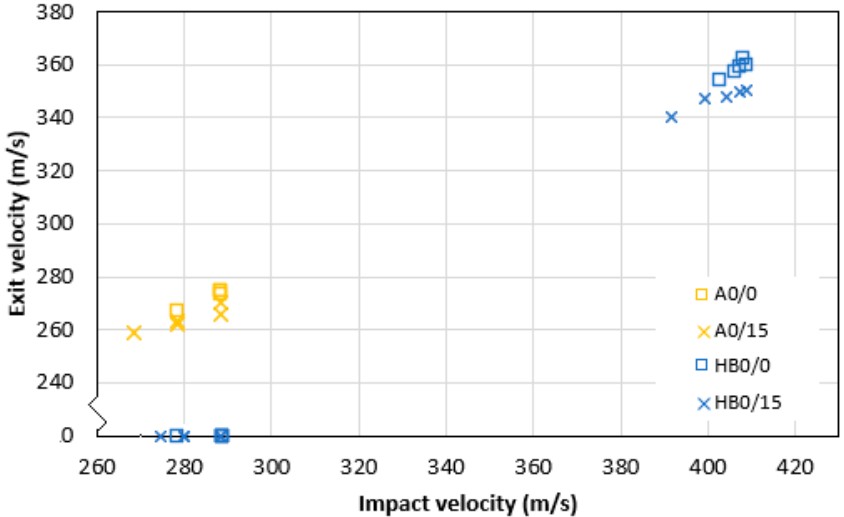

**Figure 7.** Impact and exit velocity results for aramid and HB24 L0/0 and L0/15 laminate, hit by the solid steel sphere (160 J).

The PRP (40 J) fragment was used (velocities between 265 and 290 m/s) to preform five shots against laminates of aramid and HB24 with different woven fabric orientations. Looking at the aramid laminates, it is possible to verify that the A0/15 presents a better impact absorption compared to the A0/0. This can be seen because, at the same entrance velocities, the fragment loses more velocity after hitting the A0/15 laminates, meaning that the angle-ply structure provides more resistance to the fragment movement, reducing its velocity and absorbing more of its energy.

Using the same fragment and impact velocities, it was not possible to perforate the HB24 laminates, so the exit velocity of the fragments were not measured. For that reason, at the same impact velocities already tested in the aramid laminates, the final velocity was 0 m/s. So, in order to be able to perforate the laminate, impact tests using higher energies (160 J) were performed in which the fragment can reach higher impact velocities, around 400 m/s. Due to these values, the E fragment was used.

It is possible to verify the impact tests with these entrance velocities because the fragment now managed to perforate the HB24 laminates. As was verified in the aramid laminates, the HB0/15 was the laminate that absorbs more energy and presents lower exit velocities compared to the ones registered in the HB0/0.

The impact resistance improves with the introduction of an angle-ply structure because, as can be verified in Figure 8, when a fragment hits a laminate with different woven fabric orientations (in an angle-ply configuration), the fragment has contact with more principal yarns arranged in other directions, when compared with a laminate with L0/0 configuration.

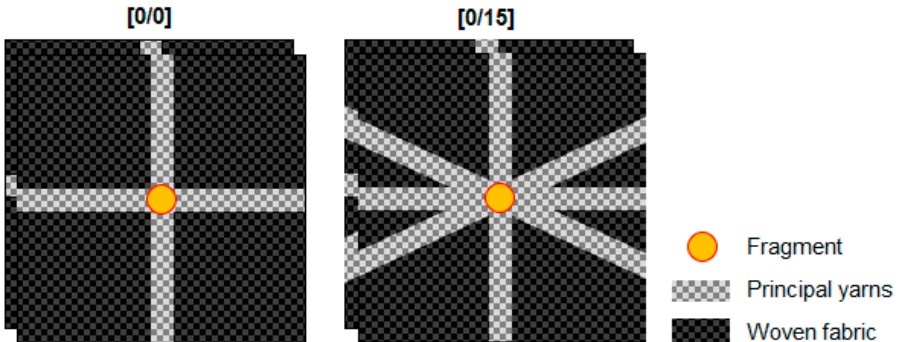

**Figure 8.** Fragment interaction with laminates with and without an angle-ply orientation.

*3.3. Impact Tests—The Effect of the Type of Fiber*

After the woven fabric orientation was verified, different laminates with L0/15 produced with different fiber fabrics, namely aramid, HB24, E-glass, and S-glass, were impact tested. These tests were carried out using the E fragment (160 J), and the measure entrance and residual velocities are presented in Figure 9.

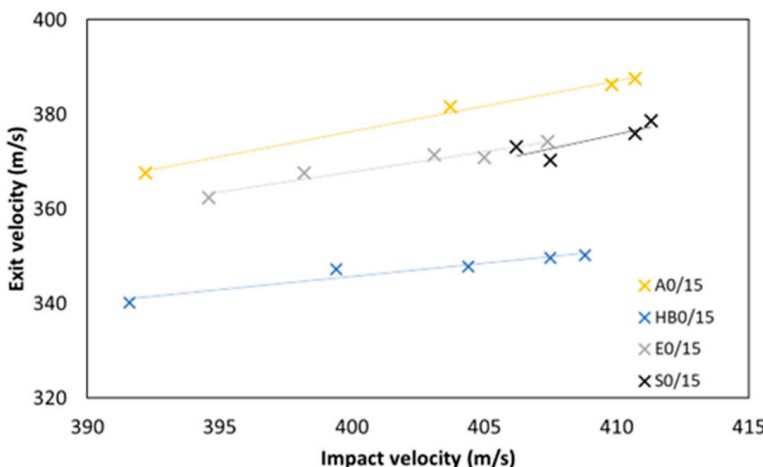

**Figure 9.** Evolution of the exit velocity as a function of the impact velocity for L0/15 laminates produced with different fiber materials and impacted by a spherical fragment animated with a kinetic energy around 160 J.

To identically fragment impact velocities, depending on the type of laminate used, different fragment exit velocities were obtained. The HB24 was the one that presents lower exit velocities; this means that laminate panels made with this material have better impact resistance properties than the other materials. In this case, for the laminates with 1 mm thickness, the aramid laminate presented the worst impact resistance because it presents higher residual fragment velocities after the impact.

Considering the E-glass and S-glass fiber laminates, they exhibit similar energy absorption, but the S-glass seems to present better results compared to the E-glass. Theoretically, these fibers have better mechanical properties than the E-glass, and, according to some available solutions in the market, the developers say that S-glass fibers present better impact properties than the E-glass [32,33]. So, the similarity of the obtained results between these glass fibers may be explained by the reduced number of woven fabric layers present in the laminate (only two). Because of this reduced number of layers and the higher areal density of the S-glass fabric, when hit by a fragment, the orientation of the woven fabrics may be not enough to present the angle-ply effect. To verify the impact properties between these two types of fibers, thicker laminates need to be produced to verify the angle-ply effect.

Figure 10 presents the entry and exit holes obtained for each laminate tested using solid steel sphere 160 J.

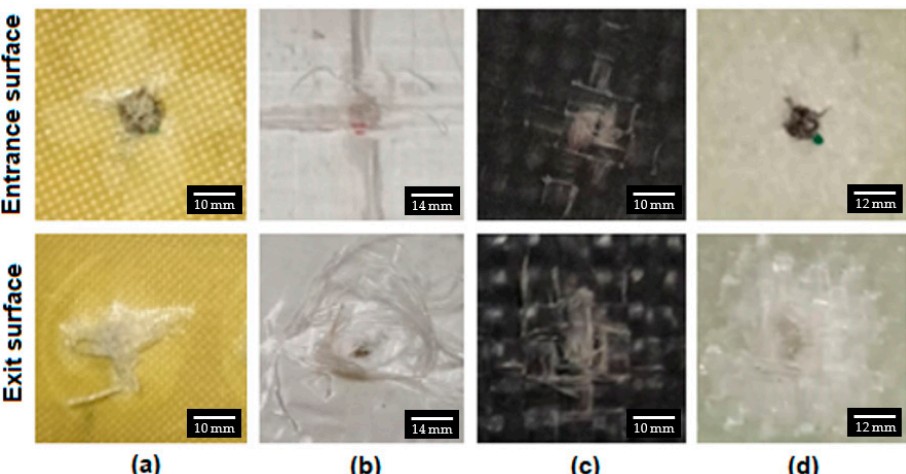

**Figure 10.** Entrance and exit surface after the solid steel sphere impact (160 J) fragment of the laminates made of (**a**) aramid, (**b**) HB24, (**c**) S-glass, and (**d**) E-glass.

The samples' surface conditions after performing the impact tests allowed us to analyze the energy dissipation mechanisms, such as the matrix cracking, the fibers rupturing, and the interface debonding and delaminating. In the HB24 laminate (with a low modulus matrix), it is possible to see the primary yarns that contact directly with the fragment. Analyzing the exit surface of the fragment (back surface), it is possible to verify the transition from the laminate planar geometry to a conical shape, promoted by the impact of the fragment, as shown in Figure 11. This behavior from the HB24 laminate shows a greater energy absorption mechanism due to the deforming during the impact, improving its capability of deforming and absorbing the impact energy for it primary and secondary fibers, forming that conical shape.

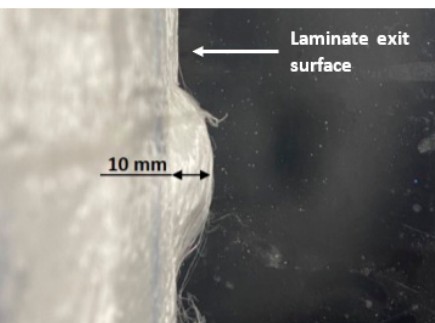

**Figure 11.** Conical shape on HB24 laminate after the ballistic impact.

On the other laminates that use a harder matrix of epoxy, in the impact surface, it is possible to verify the damage area correspondent to the deformation of the secondary yarns. These yarns were subjected to forces during the impact, deforming and showing the impact energy dispersion during the fragment impact.

Due to the capability of deforming during the fragment impact of the HB24 laminate, composed with high modulus fibers but with a low modulus matrix, this material shows greater ballistic resistance compared to the other fibers' laminates that were more rigid due to the type of fiber and matrix used, being penetrated more easily, by breaking the high modulus fibers and epoxy matrix.

## 4. Conclusions

In this study, composite laminates were developed using different woven fabrics (W0, W90) of aramid, E-glass, and S-glass woven fabrics, impregnated with epoxy resin, and UHMWPE prepregs (HB24), to produce angle-ply structures with different layer orientations, L0/0 and L0/15. The developed laminates were impact tested using a gas gun and different fragments, associated with different impact energies. The impact tests enabled us to study the effect of fragment geometry and shape; study the angle-ply effect using laminates with different layers orientation; and, finally, verify each fiber's laminates that present a better impact performance.

It was verified that higher fragment impact velocities imply higher residual velocities. For the entrance velocities used in these tests, between 120 and 420 m/s, the fragment geometry/shape presents a good linearity between the impact and exit velocities after the fragment impact.

From the angle-ply effect on the impact absorption of laminates of aramid and HB24, it was concluded that laminates with different woven fibers orientation, L0/15, present better impact absorptions compared to the L0/0, presenting lesser residual velocities for the same entrance velocities.

The HB24 fiber laminates presented a higher impact resistance, and the aramid ones presented a lower impact resistance. Between the E and S-glass fibers, it was not possible to conclude which one was better, demonstrating a need to produce thicker laminates to evidence the angle-ply effect.

According to the damage caused to the tested laminates by the fragment impacts, some of the energy dissipation mechanisms, such as the matrix's cracking, the fibers' rupture, and the interface's debonding and delamination, were verified.

Overall, the angle-ply UHMWPE laminates with the L0/15 orientation are an excellent option to place inside building walls, as they present excellent impact resistance against fragments. This type of material also has the advantage of being light (low density) and resistant to corrosion.

**Author Contributions:** Conceptualization, D.B.; Methodology, C.M.; Investigation, D.B.; Writing—original draft, D.B.; Writing—review & editing, D.B., J.B. and P.R.; Visualization, P.R.; Supervision, C.M. and J.B.; Project administration, F.C. and R.F.; Funding acquisition, R.F. All authors have read and agreed to the published version of the manuscript.

**Funding:** This research received no external funding.

**Data Availability Statement:** Authors can confirm that all data generated or analyzed during this study are included in this published article.

**Acknowledgments:** The authors would like to express appreciation for the funding provided by the Operational Program of Portugal 2020, under the project n° POCI-01-0247-FEDER-045231, entitled as "Ballistic Composite Panel".

**Conflicts of Interest:** The authors declare no conflict of interest.

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
