# Peer review of "Blast Fragment Impact of Angle-Ply Composite Structures for Buildings Wall Protection"

_buildings, doi:10.3390/buildings13081959_

Round 1

Reviewer 1 Report

1. Section 2.1 is duplicated.

2. It is difficult to obtain the difference of the impact velocity and the residual velocity in Figure 5 and Figure 6.

3. The size of projectiles should be added in the table and the descriptions of the front face and back face should be same in table 3.

4. The conclusions should be  refined.

The English language of the manuscript  is easy to understand. 

Author Response

Thank you for all the comments made. 
We check your notes and we made the following changes:

  • Delete the repeated 2.1 section;
  • Add some "internal lines" to the graphs of the Figure 5 and 7, in order to better relate the xx and yy axis;
  • Add the projectile dimension and entrande areal damage created during the impact;
  • Tried to simplify the conclusion to become more objective and understandable

Reviewer 2 Report

The article evaluated the characteristics of composites made from various fibrous fabrics and using different orientation of the layers, which are used to strengthen buildings. The topic is very relevant and useful. The abstract is well written, complete and concise in various aspects. The keywords are complete and appropriate. The introduction is well written and finished. Materials and methods are clear and well explained. The results are easy to understand and comprehensive. All studied characteristics were presented in the figures. The conclusions are written clearly and understandably. The bibliography is formatted in accordance with the requirements of the journal and does not contain inappropriate references. But there are comments on the article:

Repeated twice section materials.

Author Response

Thank you in advance for the review and for all the comments made.
We verify that we have repeated section 2.1, which we have already corrected.

Now we will send the document again, with the changes made according to the comments of all reviewers.